# Assessment of Glial Activation Response in the Progress of Natural Scrapie after Chronic Dexamethasone Treatment

**DOI:** 10.3390/ijms21093231

**Published:** 2020-05-02

**Authors:** Isabel M. Guijarro, Moisés Garcés, Pol Andrés-Benito, Belén Marín, Alicia Otero, Tomás Barrio, Margarita Carmona, Isidro Ferrer, Juan J. Badiola, Marta Monzón

**Affiliations:** 1Research Centre for Encephalopathies and Transmissible Emerging Diseases—Institute for Health Research Aragón (IIS), University of Zaragoza, C/Miguel Servet 155, 50013 Zaragoza, Spain; isabelmariagt91@gmail.com (I.M.G.); moisesgarces1@gmail.com (M.G.); belenm@unizar.es (B.M.); aliotgar@hotmail.com (A.O.); tbarrio_sc@hotmail.com (T.B.); badiola@unizar.es (J.J.B.); 2Bellvitge Biomedical Research Institute (IDIBELL), L’Hospitalet de Llobregat, 08908 Barcelona, Spain; pol.andres.benito@gmail.com (P.A.-B.); mcarmona@idibell.cat (M.C.); 8082ifa@gmail.com (I.F.)

**Keywords:** scrapie, dexamethasone, neuroinflammation, astrocytes, microglia, prion diseases

## Abstract

Neuroinflammation has been correlated with the progress of neurodegeneration in many neuropathologies. Although glial cells have traditionally been considered to be protective, the concept of them as neurotoxic cells has recently emerged. Thus, a major unsolved question is the exact role of astroglia and microglia in neurodegenerative disorders. On the other hand, it is well known that glucocorticoids are the first choice to regulate inflammation and, consequently, neuroglial inflammatory activity. The objective of this study was to determine how chronic dexamethasone treatment influences the host immune response and to characterize the beneficial or detrimental role of glial cells. To date, this has not been examined using a natural neurodegenerative model of scrapie. With this aim, immunohistochemical expression of glial markers, prion protein accumulation, histopathological lesions and clinical evolution were compared with those in a control group. The results demonstrated how the complex interaction between glial populations failed to compensate for brain damage in natural conditions, emphasizing the need for using natural models. Additionally, the data showed that modulation of neuroinflammation by anti-inflammatory drugs might become a research focus as a potential therapeutic target for prion diseases, similar to that considered previously for other neurodegenerative disorders classified as prion-like diseases.

## 1. Introduction

Prion diseases are a group of fatal neurodegenerative diseases affecting animal and human species and are caused by the conversion of cellular prion protein (PrP^c^) into a pathological isoform called pathological prion protein (PrP^sc^). Specifically, scrapie is the archetype of all these disorders. In spite of the importance of this group of diseases in public health and their occurrence as endemic disorders in many countries worldwide, studies have mostly focused on experimental instead of natural models [1,2,3,4]. However, evidence of failure in reliability, mainly in neuroscience research, is increasingly being published [5,6]. Therefore, there is growing interest in studying natural models, especially of neurodegenerative disorders, such as prion diseases.

Neuropathological hallmarks of prion diseases include spongiform changes, vacuolation, astrogliosis, microglial activation, neuronal loss and accumulation of PrP^sc^ [7,8,9]. Several lines of research have found that this group of diseases shares neuropathological features, extracellular deposit of insoluble plaques of an aberrant protein, astrocytosis and microglial activation [10,11,12] with other human neurodegenerative disorders. Moreover, the mechanism of induction and spread of protein misfolding within this group of diseases have led to include them in prion-like disorders [13], such as Alzheimer’s (AD), Parkinson’s (PD) and Huntington’s (HD) diseases. In line with this fact, some studies have proposed using prion diseases as acceptable models to better understand the pathogenesis of prion-like disorders [14,15].

Currently, neuroinflammation is considered as an intrinsic characteristic of all the above-mentioned neuropathologies [16]. Indeed, the first description of an innate inflammatory response in a neurodegenerative process was made 25 years ago in AD [17], and subsequent studies have also found inflammatory components in other prion-like diseases [18] as well as prion diseases [19,20,21]. Neuroinflammation has been correlated with the progress of neurodegeneration [16] and is even being proposed as a pathogenic mechanism in disorders such as AD [22], supporting the hypothesis that neuroglia constitute potential neurotoxic cell populations [23,24,25,26,27,28,29]. The neuroinflammatory process is defined as the prolonged activation of microglial cells with the consequent production of pro-inflammatory cytokines [30]. Moreover, despite the fact that this glial cell type has gained more attention than astrocytes in this process, astroglia have also been demonstrated to be highly involved in scrapie [31,32,33,34], human prion diseases [35,36,37] and prion-like [38,39,40] diseases.

Nevertheless, the notion of neuroinflammation being a protective response against neurodegeneration is also supported by other authors. The inflammatory process appearing in several neurodegenerative diseases has been attributed to a protective role for both astroglia [39,41] and microglia [42,43,44,45]. Therefore, a major unanswered question is the exact role played by glial cells (which constitute innate immune cells in the central nervous system, CNS) in neurodegenerative disorders.

Synthetic glucocorticoids (GCs) have been used therapeutically in several inflammatory disorders. Consequently, they would be the first choice to control neuroinflammation. In fact, these synthetic hormones have been demonstrated to represent the main regulators of neuroglial inflammatory activity [46], resulting in clinical benefit in AD [47]. In the same manner as for AD or PD, modulation or inhibition of neuroinflammation might be a therapeutic target for prion diseases [48]. To our knowledge, some studies based on corticosteroid treatments were tested in scrapie many years ago, but the main focus was on targeting PrP^sc^ or PrP^c^ [49,50,51] and not the neuroinflammatory process.

The specific objective of this study was to assess the effect of the synthetic GC dexamethasone (DEX) on the spread of scrapie in a natural sheep model, paying special attention to the differential expression of astroglial and microglial markers as main components of the host immune response in the brain. 

The overall goal consisted of determining the beneficial or detrimental role of these glial cells in the neurodegenerative progress of prion diseases. Since the natural model of scrapie (archetype of prion diseases) was used in the present study, it might be a feasible and reliable model to extrapolate results to prion-like diseases.

## 2. Results

### 2.1. Clinical Signs

Clinical sheep, regardless of whether they were treated with DEX or not, showed motor symptoms (tremors, ataxia, incoordination and prostration) likely associated with deep affectation of the cerebellum, as well as pruritus in most cases. Table 1 shows the main clinical signs developed by each clinical sheep during the experiment. Control sheep did not present any of the key features of scrapie throughout the time of the experiment in any cases.

As expected, DEX-treated sheep (mainly from the control group, due to the extended duration of treatment) experienced lesions compatible with secondary effects of the GC, including Cushing’s syndrome, in 3 out of 4 cases. Nevertheless, wound lesions were the major safety concern due to the long-term use of DEX (3 out of 4 control-treated sheep were affected by extensive alopecia).

Regarding survival time, although no statistically significant differences were observed between Kaplan ¬–Meier curves in clinical sheep, one of the ten (10%) treated clinical sheep survived 155 days compared to 72 days as the maximum observed for the non-treated clinical animals (Figure 1).

### 2.2. Histopathological Findings (H-E)

Spongiosis was absent in all regions examined in sheep samples of both treated and non-treated control groups, while it was widespread in all areas in the clinical groups. Vacuolation was mainly located in the neuropil in all cases where it was found, although intraneuronal vacuoles were also observed in some cases. In regard to brain regions, subtle spongiform changes were found in the Fc, while they were much more pronounced and severe in the MO of all clinical animals. Thus, caudal tissues were the most affected by spongiform changes (Figure 2A). Regarding the specific impact of DEX administration on spongiform changes, no significant differences were found between treated and non-treated clinical sheep (*p* > 0.05) (Figure 2B). 

The significant decrease in motor activity observed in treated and non-treated clinical sheep was consistent with cell damage observed in the Purkinje cell layer, which presented severe vacuolation and even partial disappearance of Purkinje cells in some cases. Furthermore, these cells appeared swollen with neurite thickening (torpedoes) (Figure 3).

### 2.3. Immunohistochemical (IHC) Findings

#### 2.3.1. PrP^sc^ Accumulation

No PrP^sc^ deposits were observed in any tissues belonging to control animals. On the other hand, prion protein deposition was widespread in all brain areas examined in both treated and non-treated clinical groups. Quantitative differences concerning PrP^sc^ deposits were found between the Cb and Fc in both groups of clinical animals (Figure 4A). No significant differences between treated and non-treated clinical animals were found by the Mann–Whitney U test (Figure 4B). 

Although lineal, spot, coalescent or granular PrP^sc^ deposition patterns could be observed on some occasions, the coalescent pattern was the most frequently observed pattern in all brain regions. Purkinje cells in the cerebellum were never immunostained for this marker.

#### 2.3.2. GFAP

As previously described, paraffin sections processed for IHC showed an increase in GFAP immunoreactivity in clinical scrapie when compared with controls (Figure 5A).

Treated control animals exhibited significantly higher immunolabeling for GFAP than the non-treated control group in all regions, except the obex (Fc, *p* < 0.05; Cb, *p* < 0.05; MO, *p* < 0.01) (Figure 5B). Strikingly, by contrast, GFAP immunostaining did not reveal major differences regarding the effect of treatment in any brain area examined in both clinical groups (Figure 5C). A multivariate linear regression analysis verified that the effect of treatment was significant (*p* < 0.05).

Morphologically, GFAP immunolabeling was found surrounding the meningeal zones in all regions examined. In the cerebellum, as previously described, the Purkinje cell layer was often immunostained, and an intense radial profile of GFAP in the molecular layer was found in samples with the highest intensity. Meanwhile, samples with the lowest GFAP intensity predominantly presented a horizontal profile in this layer of the cerebellum. In addition, the vast majority of astrocytes in the medulla oblongata and obex showed hypertrophic morphology in treated controls compared to astrocytes in the non-treated sheep, which demonstrated their typical stellate morphology (Figure 6).

#### 2.3.3. IBA-1

As expected, the IHC technique demonstrated an expansion of microglial populations (IBA-1^+^ cells) in clinical animals compared with controls. However, no differences were found regarding the intensity of microgliosis among brain regions either for the control or clinical group (Figure 7A).

Contrary to those observed changes for astroglia in controls, the microglia immunostaining pattern did not significantly change upon DEX treatment during clinical or control stages. Thus, no statistically significant differences in intensity were found between microglia in treated and non-treated animals (Figure 7B,C).

Morphologically, the Purkinje cell layer in the cerebellum was never immunostained for IBA-1, and the ramified phenotype was the most frequently observed morphology. Additionally, a relevant higher percentage of microglial cells in this encephalic area presented an amoeboid phenotype in treated controls compared to microglia in non-treated sheep (Figure 8).

#### 2.3.4. RT-qPCR

Although it was not statistically significant (*p* = 0.064), daily injections of DEX in clinical sheep tended to show increased GFAP mRNA levels in the cerebellum compared with non-treated clinical animals. However, no significant differences were found in the mRNA expression of the microglial markers tested (AIF-1 and CD68 genes) in the cerebellum (Figure 9A).

Regarding the frontal cortex, AIF-1 (or IBA-1) expression tended to be lower in clinical DEX-treated sheep compared with the non-treated group, although this was not statistically significant (*p* = 0.085). No changes were detected for astrocytic markers (GFAP and ALDH1L1) in this brain region (Figure 9B).

## 3. Discussion

Currently, no treatments for prion diseases exist. To our knowledge, most of the few studies testing potential treatments in scrapie as an archetype of this group of disorders were carried out many years ago. Moreover, their main focus was on targeting PrP^sc^ or PrP^c^ instead of the process of neuroinflammation. Here, the effect of long-term GC administration on survival period, PrP^sc^ deposition, spongiosis and glial response was evaluated using naturally infected scrapie sheep as a natural model of neurodegenerative diseases. Thus, the purpose of this experiment was to determine how the synthetic GC DEX influences the host immune response in order to characterize the beneficial or detrimental role of glial cells in the progress of neurodegeneration of prion diseases. Previous studies testing different strategies to manipulate microglial function reported contrary effects on survival period in experimental models of prion diseases [29,45]. This prompted us to more deeply investigate the role of host immunity during the course of natural infection.

Intensive research efforts have been made to look for a therapeutic solution for neurodegenerative diseases based on immunomodulation. The development of therapeutic strategies that inhibit NF-kB activity was proposed to tackle these pathologies [52]. In fact, DEX had been used as a candidate glial cell modulator [53] and was found to protect neurons via decreased neuroinflammation of glial cells in a mouse model of PD [54]. More recently, GC therapy demonstrated clinical benefit in AD [47] and was associated with a lower risk of dementia in humans [55]. In contrast, GCs were described as neurotoxic for AD in murine models since they enhanced the pathology, augmenting deposits of amyloid beta and tau [56]. In regard to prion diseases, Outram et al., (1974) observed a reduction in the susceptibility to murine scrapie infection of mice treated with GCs [57]. Treatment with ibuprofen in murine scrapie was not conclusive due to the risky side effects [58].

The results described here regarding scrapie at clinical stages of disease partially agree with previous reports showing that no treatment is effective when neuronal degeneration has already begun. This is consistent with descriptions for non-steroidal anti-inflammatory drugs (NSAIDs) in AD [59], suggesting that prevention strategies are necessary instead of treatments. No beneficial long-term effect on disease progression was demonstrated for GC therapy in multiple sclerosis (MS) [60]. However, it is worth mentioning that, in the present study, one clinical sheep extended its survival period after DEX treatment. Even though this was only observed for one individual, it represents an encouraging result, as it opens up the possibility for anti-inflammatory therapy to have potential in at least some cases. Further studies to understand this potentiality are being designed.

Of note, corticosteroids have been established to cause dose-related immunosuppression, yet the mechanisms behind this impaired immune function have not been defined [61]. GCs exhibit paradoxical immunomodulatory functions [62,63]; although traditionally known as anti-inflammatory drugs, sometimes the modulating mechanisms fail, and they can aggravate CNS inflammation [64]. The paradoxical effects of GCs on neuronal survival and death have been attributed to the concentration and the ratio of receptor activation. GC-induced leucine zipper (GILZ) is a recently identified protein transcriptionally upregulated by GCs. Constitutively expressed in many tissues including the brain, GILZ mediates many of the actions of GCs. It mimics the anti-inflammatory and antiproliferative effects of GCs, but also exerts differential effects on stem cell differentiation and lineage development. Together with the dosage of GCs, the length of treatment in relation to the immune response is decisive to determine if GCs exhibit pro- or anti-inflammatory properties [65]. Acquired resistance is another problem, according to descriptions in MS after treatment for 3–6 months [66]. All these reasons could explain the lack of differences observed in this study regarding the length of treatment, despite varying from some weeks to nearly 18 months.

Although some studies argue that DEX has minimal access to the CNS [67,68,69], the present study clearly demonstrates the successful efficacy of DEX to cross the blood–brain barrier, as evidenced by the strong astroglial reaction in treated controls. However, this experiment also presented further complications due to the issue of GC therapy itself. DEX has a number of adverse effects, mostly associated with suppression of immunity [70]. The same side effects as those described by other authors [71,72] were observed in the present study, with outstanding wound loss as the most frequent and highly adverse side effect. On many occasions, animals needed to be euthanized because of this effect. For this reason, it was difficult to conclude whether this immunosuppressive therapy might ameliorate clinical signs or slow down the neurodegenerative process. The data provided herein should be interpreted with caution due to the use of a natural model, along with the inherent difficulties described above.

Regarding the specific impact of DEX administration on neuropathological lesions, no significant differences in vacuolation or PrP^sc^ deposition were found between treated and non-treated clinical sheep. The observed cell damage in the Purkinje cell layer of the cerebellum is in agreement with previous findings in scrapie [33], Creutzfeldt–Jakob disease (CJD) [36] and other neurodegenerative disorders [73]. When these cells were closely observed in ultrastructural studies, the vacuolation occurring around this cell type displayed a close relationship with glial cells [74]. In regard to differences among brain regions, the caudal areas were the most affected by spongiform changes and PrP^sc^ deposits. They were more frequent and widespread in the cerebellum compared to the frontal cortex, as previously reported in scrapie [75]. PrP^sc^ presented different deposit patterns, although the coalescent pattern was most frequently observed in all brain regions, as was previously found in the cerebellum from CJD-affected individuals [36]. In contrast, a recent study demonstrated that PrP^sc^ accumulates in all brain regions independently of neurodegeneration [76]; this discrepancy in results might be due to differences in models, since the study used a mouse experimental model of Gerstmann–Sträusler–Scheinker syndrome, and the other researchers used natural models of scrapie and CJD. This reinforces that we should be cautious with extrapolation from experimental models to reality.

Focusing on glial cells, which are the key topics in this study, both activated astrocytes and microglia, based on morphology as well as specific marker expression, were observed in clinical animals compared to controls. The activation of glia in scrapie [32,33] and other prion-like diseases [40,77] has been exhaustively characterized. However, this has not previously been described as a result of anti-inflammatory treatment.

In this study, a statistically significant difference in GFAP marker expression was observed in all regions examined, except the obex, after DEX treatment compared to controls. By contrast, treatment did not reveal major differences between clinical animals. This may be due to downregulation of astrogliosis, reflecting astroglial paralysis in the clinical stage of scrapie, as was recently described in late stages of AD [78,79]. Further studies would be essential to confirm this assertion.

The cerebellum is a preferential target of prions in scrapie [26,80] and CJD [81,82,83]. In fact, a relevant finding in this encephalic area could help to clarify astrocytic behavior in the neurodegenerative progress of prion diseases. An intense radial profile was observed in the molecular layer with high intensity, while the horizontal profile was instead related to samples with low intensity of this marker. This is the same as that evidenced in our previous work for human prion and prion-like disease samples [36,40]. As claimed then, this pattern suggested a possible glial stem cell response in order to protect against or compensate for neuronal loss [84]. This would agree with the hypothesis about astroglial paralysis, which, despite trying to react against brain damage, might prevent competent astrocytes from being formed. In the same vein, RT-qPCR results demonstrated a tendency toward increased GFAP mRNA, which is consistent with this assumption. Even though astroglia seem to attempt to compensate for the damage by initiating proliferation/regeneration, an error somewhere in this process could be blocking the final aim.

Another GFAP morphological finding was that the vast majority of astrocytes presented a hypertrophic morphology in treated control samples (similar to that at the clinical stage) in comparison with the typical stellate form in non-treated samples. It is well known that microglia undergo complex metamorphoses when they are reactive. However, an enigma of control of astroglial morphology in brain physiology is beginning to emerge [85]. Indeed, this finding is, to our knowledge, the first description of this phenomenon in astroglia.

In this in vivo study, comparing non-treated clinical and control samples, there was also an expansion of the microglial population in the clinical stage, as expected. This resulted in an increased number of microglia associated with an activated and phagocytic phenotype, as previously reported [86,87]. Nevertheless, microglial immunostaining intensity did not significantly change after DEX treatment (neither in the clinical nor control group). Similarly, a previous study [88] did not observe a decrease in microglial activation after lipopolysaccharide (LPS) induction of neuroinflammation and intranasal DEX treatment in mice, suggesting that the dose was not sufficient to reduce IBA-1 expression. However, other studies managed to reduce microglia activation with DEX sodium phosphate by using cell cultures [89]. The discrepancies in results may be due to different experimental models, pharmacological presentation or routes of DEX administration. Indeed, recent studies have shown that the effects of GCs on brain inflammatory responses are truly complex [90].

Concerning morphological findings, a high percentage of microglial cells in the cerebellum presented an amoeboid phenotype in treated controls compared to non-treated ones, which appeared to be more ramified. Provided that the amoeboid phenotype represents the most activated microglial shape [91,92,93] associated with the expression of neuroinflammatory genes [94] and the presence of this amoeboid phenotype in prion diseases seems to be stimulated by the high accumulation of PrP^sc^ [95], we might speculate that DEX treatment stimulates phagocytosis of PrP^sc^ deposits, which would constitute a useful tool against prion progress. However, this stimulation was not evident in clinical animals despite the same treatment administration. This is consistent with the idea of a failure of the glial response.

Regarding gene expression analysis, DEX treatment in clinical sheep involved an increase in GFAP and a decrease in AIF-1 (also known as IBA-1) mRNA expression in the cerebellum and frontal cortex, respectively. This finding could be related to the previously described region-specific pattern of neuroinflammation in sporadic CJD [96], in accordance with different cytokine profiles found in these two brain regions. It is worth mentioning that glia immunostaining and its respective mRNA expression were not correlated in recently reported previous studies [88,97], an aspect that needs further clarification. Nevertheless, in this study, the tendencies demonstrated by high-resolution molecular analysis confirmed the overall results provided by IHC analysis. Consequently, both techniques, which herein were demonstrated to complement one another, led to the conclusion of a probable glial failure.

It is currently assumed that glial responses can play both protector and toxic roles depending on the degree of activation [37,98,99], supporting the concept that neuroinflammation induced by glia can amplify pathology [24,25]. The transition from neuroprotective to neurotoxic activity of astrocytes by cytokine stimulation had been demonstrated [53,100,101], but such neurotoxicity was prevented when astrocytes were treated with DEX in cell culture, while DEX had no effect on neurons. On the basis of these observations, DEX could promote neuroprotective properties of astrocytes, as confirmed here in the natural model of prion disease. Nevertheless, the findings presented in this study support a potential failure of astrocytes and not a role for enhancing pathology.

A recent study reported that neurotoxic astrocytes (called A1) are potential contributors to neuronal death in several neurodegenerative disorders [102,103]. This subtype of astrocytes might be involved in damaging actions, although the interaction of both populations, astroglia and microglia, has been postulated as a candidate involved in neurodegeneration [103,104], resulting in the essential presence of microglia [104]. As a consequence, drugs to block the release of neurotoxins by these astrocytes have been suggested as a possible solution [103]. Currently, the activation of astrocytes remains poorly understood. The release of cytokines from those astrocytes and microglia accompanies this event. Provided that the present study was focused on glial cells, the next goal would consist of providing new information about the mechanisms underlying cytokine release. As has been previously suggested, it could be a crucial target for therapeutic approaches in CNS in prion diseases [101]. It is indispensable to study how glial communication might prevent neuronal damage.

Distinct conformations of PrP^sc^ might explain the unusual wide range of neuropathological, biochemical and clinical features of prion diseases [105] and may contribute to this disagreement in conclusions regarding treatments. Since a natural model was used in this study, it is likely that natural Transmissible Spongiform Encephalopathies (TSE) infections of ruminants involve mixtures of strains rather than a single strain [106], but it is necessary to emphasize that it would reflect the reality much better.

In conclusion, we would like to reassert the essential requirement of using natural models to provide reliable results. It constitutes a powerful in vivo approach to assess the activation of glial cells by anti-inflammatory therapy in a trustworthy and quantitative manner. However, it is also much more time consuming and entails inherent difficulties and uncontrollable factors, mainly the exact time of natural infection in the field. Transgenic models have demonstrated poor representativeness of natural disease progress; some examples include studies that sought to delay the progress of amyotrophic lateral sclerosis (ALS) in a murine model [107]. Unfortunately, anti-inflammatory treatment worsened symptoms in clinic phase III in ALS-affected humans [6] or in epidemiological studies with drugs that managed to reduce the risk of developing PD and AD [108], while clinical assays in sick patients failed [5].

Taking into consideration the overall results presented in this study and that some authors postulated that anti-inflammatory therapeutic approaches could be combined with other strategies to achieve improved therapeutic effects [109], combining this strategy with a natural model, such as that described in the present study, may confer an appropriate starting point to advance the subject.

## 4. Material and Methods

All the following experimental procedures were approved by the Ethical Committee of the University of Zaragoza (Comisión Ética Asesora para la experimentación animal, 28 September 2016; Reference number, PI41/16). All efforts were made to minimize animal suffering during the experiments and to reduce the number of animals used.

The experiments were performed on 10 healthy and 15 clinical scrapie Rasa Aragonesa ewes. Affected animals belonged to positive flocks of scrapie in Zaragoza (Spain) and presented clinical signs, and their status was confirmed by the presence of pathological prion protein in recto-anal mucosa-associated lymphoid tissue (RAMALT) biopsies (Figure 10). Healthy sheep belonged to negative flocks, where no Scrapie cases had ever been detected and the absence of pathological prion protein by RAMALT biopsies as well as the absence of clinical signs were confirmed. Their age ranged from 4 to 10 years and all of the animals presented the ARQ/ARQ genotype, except for one animal with ARQ/ARH. All of them were housed in two independent groups (control and clinical) in aired and illuminated rooms with free access to daily concentrate plus food and water.

The clinical diagnosis of scrapie animals for inclusion in the study was based on classical signs, such as pruritus, tremor, locomotor incoordination and behavioral changes. Healthy sheep included in the study did not show clinical signs before inclusion and throughout the experiment.

It was considered crucial to include control animals in order to assess the effect of DEX treatment on healthy individuals because it had been not previously tested in ovine species. In the present study, 4 out of 10 controls and 10 out of 15 scrapie-affected sheep at clinical stage were treated. A summary of cases is shown in Table 2.

Treated sheep (control and clinical) were intramuscularly injected daily with DEX (SYVA, León, Spain; 0.04 mg/kg) in alternating posterior limb after a one-week period of acclimation and until euthanasia by endpoint criteria (16 months in the longest case). Non-treated animals (control and clinical) were injected in the same place and under the same conditions, but with physiologic saline solution (SSF), and endpoint in untreated controls was established after the completion of the experiment, since all animals of this group (but two, due to a second pathology) were alive. In addition, a daily dose of 0.5 mg/kg of omeprazole was administered to all sheep in order to avoid the appearance of gastric ulcers that GCs can cause [110]. Clinical animals were monitored once per day for the development of clinical signs of the disease, which typically included behavioral changes (fixed stare, isolation and hyperexcitability), trembling, weight loss or emaciation, pruritus (main symptom in sheep, often leading to wool loss) and impaired vision [111,112,113]. The appearance of clinical signs unavoidably led to prostration, which was defined as a humane endpoint requiring euthanasia in both treated and non-treated clinical groups.

As expected, DEX-treated sheep (mainly from the control group, due to the extended duration of treatment) experienced lesions compatible with secondary effects of the GC, including Cushing’s syndrome. Nevertheless, wound lesions were the major safety concern due to the long-term use of DEX (3 out of 4 control-treated sheep were affected by extensive alopecia). Due to this fact, they had to be euthanized by endpoint criteria.

Following intravenous pentobarbital injection and exsanguination, necropsy of each animal was performed. A total of 80 samples were collected and distributed for different studies. One hemisection from each sample was fixed by immersion in 4% paraformaldehyde for histopathological and immunohistochemical studies and the other hemisection was frozen at –80 °C for RT-qPCR studies. *Postmortem* interval between death and tissue processing was not less than 1 h.

### 4.1. Histopathological Studies (H-E)

Hematoxylin and eosin (H-E) staining was performed on paraffin-embedded 4 μm sections in order to visualize the neuropathological lesions in different brain areas, namely, medulla oblongata (MO), obex (O), cerebellum (Cb) and frontal cortex (Fc). Spongiosis was assessed by counting the number of vacuoles present in the grey matter from each section and scored from 0 (minimum) to 4 (maximum) by two independent observers, as previously published by the group [36,40].

### 4.2. Immunohistochemical (IHC) Techniques 

Immunohistochemistry was carried out to assess the accumulation of PrP^sc^, astrogliosis and microglial activation in the different brain areas selected. 

After specific pretreatments for antigen retrieval, immunohistochemical protocols by using specific primary antibodies against PrP^sc^ and glial markers were applied. EnVision system (DAKO, Glostrup, Denmark) and diaminobenzidine (DAB; DAKO, Glostrup, Denmark) were used as the visualization system and chromogen, respectively. Hematoxylin counterstaining and mounting in DPX were performed on all sections. Table 3 summarizes all primary antibodies and protocols used.

As previously described [36,40], all slides were assessed by two independent operators, who scored the intensity of PrP^sc^ accumulation and gliosis from 0 (absence) to 4 (maximum). Glial morphology in 10 microscopic fields in each brain region was also evaluated.

#### 4.2.1. PrP^sc^ Detection

As previously published [114], 98% formic acid immersion for 15 min, proteinase K (4 μg/mL; Roche, Reinach, Switzerland) treatment for 15 min at 37 °C and hydrated heating for 20 min preceded endogenous peroxidase blocking (DAKO, Glostrup, Denmark) for 5 min and incubation with the monoclonal antibody L42 (1:500, 30 min RT; DAKO, Glostrup, Denmark).

#### 4.2.2. Glial Fibrillary Acidic Protein (GFAP) Detection for Astrogliosis

After endogenous peroxidase blocking (DAKO, Glostrup, Denmark) for 5 min, slides were incubated with a polyclonal antibody against glial fibrillary acidic protein (GFAP, 1:500, 30 min RT; DAKO, Glostrup, Denmark).

#### 4.2.3. Ionized Calcium-Binding Adaptor Molecule-1 (IBA-1) Detection for Microgliosis

Heat treatment for 20 min was necessary before endogenous peroxidase blocking (DAKO, Glostrup, Denmark) for 5 min. Afterward, sections were incubated with a polyclonal antibody against ionized calcium binding adaptor molecule-1 (IBA-1, also known as allograft inflammatory factor-1 (AIF-1), 1:1000, overnight 4 °C; WAKO, USA).

#### 4.2.4. RT-qPCR

Cerebellum and frontal cortex frozen tissues from treated and non-treated clinical sheep were included in the following comparative molecular analysis for some glial markers.

#### 4.2.5. RNA Purification

RNA purification was performed following supplier’s instructions (RNeasy Lipid Tissue Mini Kit; Qiagen, GmbH, Hilden, Germany). RNA integrity and 28S/18S ratios were determined with the Agilent Bioanalyzer (Agilent Technologies Inc, Santa Clara, CA, USA). Samples were subjected to DNase digestion, and RNA concentration was evaluated using a NanoDrop spectrophotometer (Thermo Fisher Scientific, Waltham, MA, USA). Only RNA samples with optical density OD 260/280 ratios close to 5.0 were selected for reverse transcription.

#### 4.2.6. Retrotranscription

RNA retrotranscription into cDNA was performed according to the manufacturer’s manual (High-Capacity cDNA Reverse Transcription Kit; Applied Biosystems, Foster City, CA, USA).

#### 4.2.7. RT-qPCR

Gene expression of the astroglial markers, GFAP and aldehyde dehydrogenase 1 family member L1 (ALDH1L1), was assessed. Similarly, gene expression of the microglial markers, allograft inflammatory factor-1 (AIF-1, also known as IBA-1) and CD68 molecule (CD68), was also assessed. The parameters of the reactions were as follows: 50 °C for 2 min, 95 °C for 10 min, 40 cycles of 95 °C for 15 sec and 60 °C for 1 min. 

Table 4 shows the TaqMan probes used for these studies. Data were assessed using the ΔΔCt method using hypoxanthine phosphoribosyl transferase-1 (HPRT-1) and β-glucuronidase (GUS-β) as reference genes.

### 4.3. Statistical Analysis

Kaplan–Meier survival curves of both treated and non-treated clinical animals were performed. Statistical differences between curves were evaluated with the log-rank test (Mantel–Cox test). 

For results provided by IHC techniques, the normality of distribution was first tested with the Kolmogorov–Smirnov test. The nonparametric Mann–Whitney U test was used to assess quantitative differences between treated and non-treated groups. Moreover, a multivariate lineal regression was performed in order to detect whether some differences in GFAP marker could be associated with treatment.

Data provided by RT-qPCR were evaluated by Student’s t-test after assessing normality with the Kolmogorov–Smirnov test.

SPSS software (SPSS Statistics for Windows, Version 17.0) was used for all analyses and significance in all cases was considered at *p* < 0.05. All graphs were performed with GraphPad Prism 6.0. Data presented in figures were expressed as means and the standard error of the mean (mean ± SEM).

All statistical analyses were advised and supervised by *Servicio de Apoyo Metodológico y Estadístico* (SAME)—IACS.

## 5. Conclusions

The therapeutic approach tested in this work directly influenced astroglial responses in healthy animals. However, no effect in the clinical stage of scrapie was demonstrated, likely due to an impaired astroglial response in affected animals. Our experimental results point to a neuroinflammation-augmenting effect of DEX in control animals. Therefore, new information was obtained in this study. The present data showed advances regarding the role of glial cells in scrapie. Although treatment did not seem to be clinically relevant to disease progress when clinical signs had already begun, the evident extension of survival in one case was hopeful.

The involvement of immune response in scrapie progression have been shown by us and other researchers. This study also confirms the occurrence of neuroinflammation in neurodegeneration, as previously described by Ransohoff [16]. DEX induces rapid degranulation of mast cells (important cells of neuroinflammatory pathogenesis), which release pro-inflammatory molecules promoting activation of microglia and astrocytes. The increase of GFAP mRNA levels and decrease of AIF-1 gene expression may show an effect of a chain reaction, where DEX induces autocrine/paracrine cell signaling in which mast cells are activated first, followed by the activation of microglia. Consequently, the activation of microglial cells was reduced, AIF-1 gene expression was decreased and astrocyte activation was increased. Further studies focusing on these mast and glial cells would provide insight into the pathogenesis of scrapie in CNS and contribute to understanding the relationship between prions and the immune system.

In the future, how GCs act on different cellular types and encephalic areas to produce such different immunomodulatory results in prion diseases should be determined. Collectively, these results reinforce the importance of performing basic research in order to better understand the unexpected capacity of GCs to enhance aspects of CNS inflammation in neurodegenerative diseases.

## Figures and Tables

**Figure 1 ijms-21-03231-f001:**
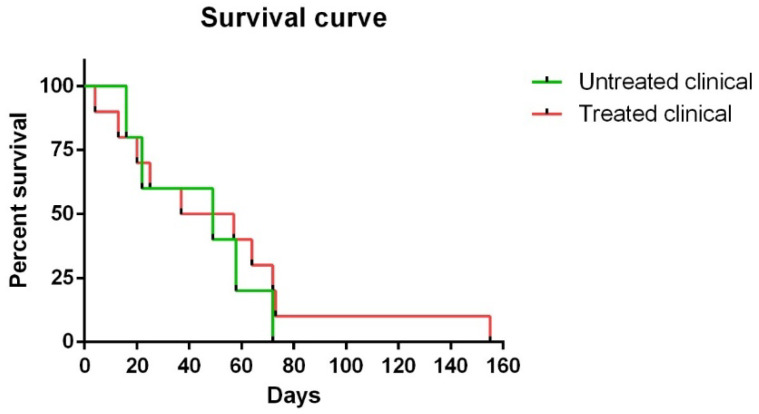
Kaplan–Meier survival curves corresponding to non-treated and treated clinical sheep. Note that one of ten treated clinical sheep survived 155 days, which was much longer than the maximum (72 days) observed for non-treated clinical animals.

**Figure 2 ijms-21-03231-f002:**
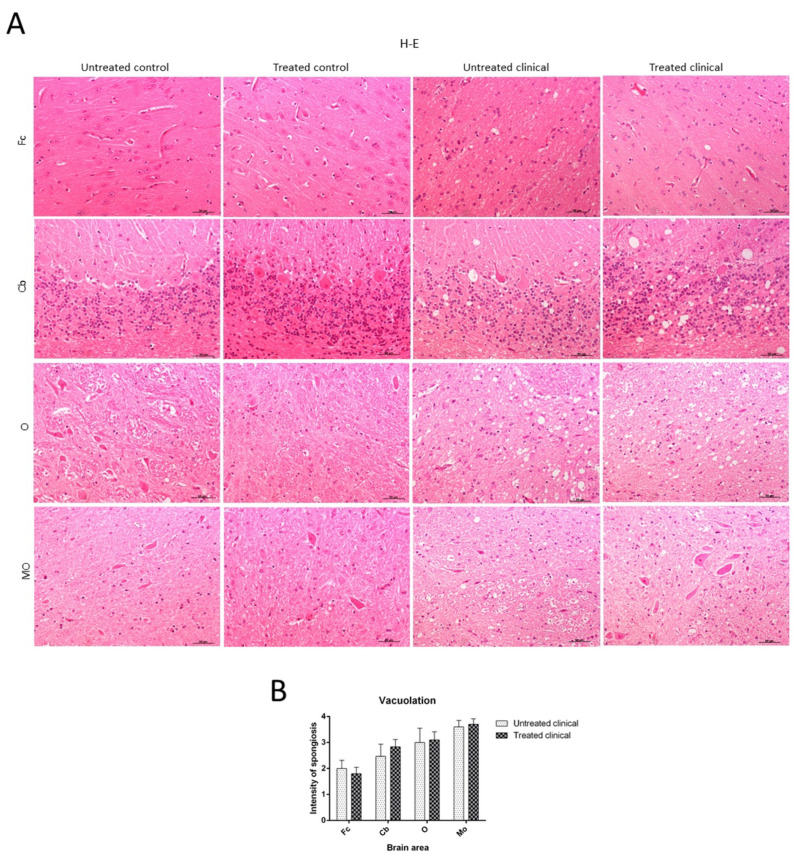
Vacuolation intensity (H-E staining). (**A**) Spongiosis was absent in all regions examined in both treated and non-treated control groups, while it was widespread in all tissues in clinical groups. Vacuolation was most frequently located in the neuropil. Note that spongiform change was most pronounced and severe in most caudal tissues. (**B**) Statistical analysis evidenced no differences between treated and non-treated clinical sheep. Medulla oblongata (MO), obex (O), cerebellum (Cb) and frontal cortex (Fc).

**Figure 3 ijms-21-03231-f003:**
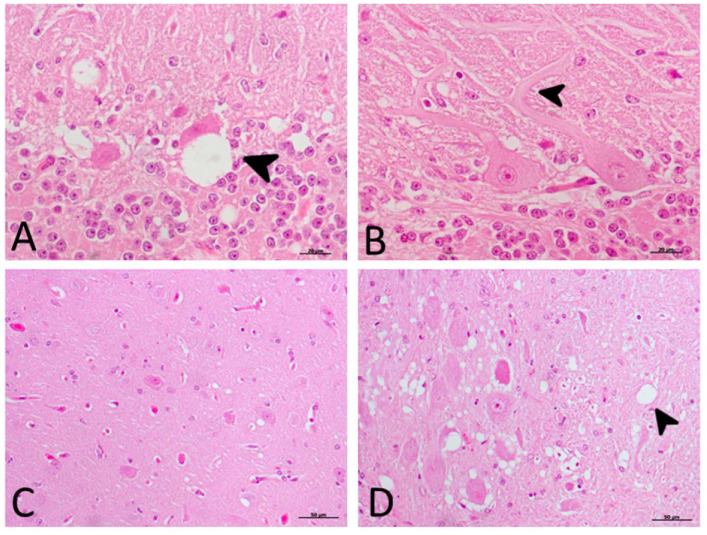
Main morphological findings in clinical sheep (H-E staining). (**A**) Cell damage observed in the Purkinje cell layer of cerebellum. Note the severe intraneuronal vacuolation (arrowhead) and partial disappearance of Purkinje neurons. (**B**) Purkinje cells appeared swollen with higher neurite thickening (torpedoes, arrowhead). (**C**) Subtle spongiform change was found in the frontal cortex. (**D**) Meanwhile, neuropil vacuolation (arrowhead) was pronounced and severe in the medulla oblongata in all cases.

**Figure 4 ijms-21-03231-f004:**
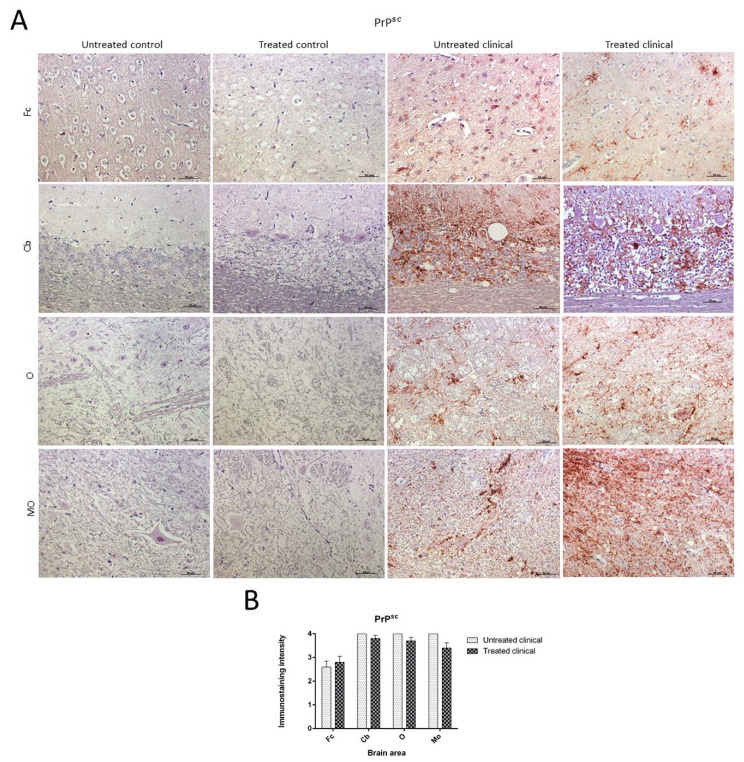
Pathological prion protein deposition by immunohistochemistry (IHC) with L42. (**A**) No PrP^sc^ deposits were observed in any tissues belonging to animals from control groups. PrP^sc^ deposition was widespread in all brain regions examined in both treated and non-treated clinical groups. (**B**) No statistical differences between treated and non-treated clinical sheep were found. Medulla oblongata (MO), obex (O), cerebellum (Cb) and frontal cortex (Fc).

**Figure 5 ijms-21-03231-f005:**
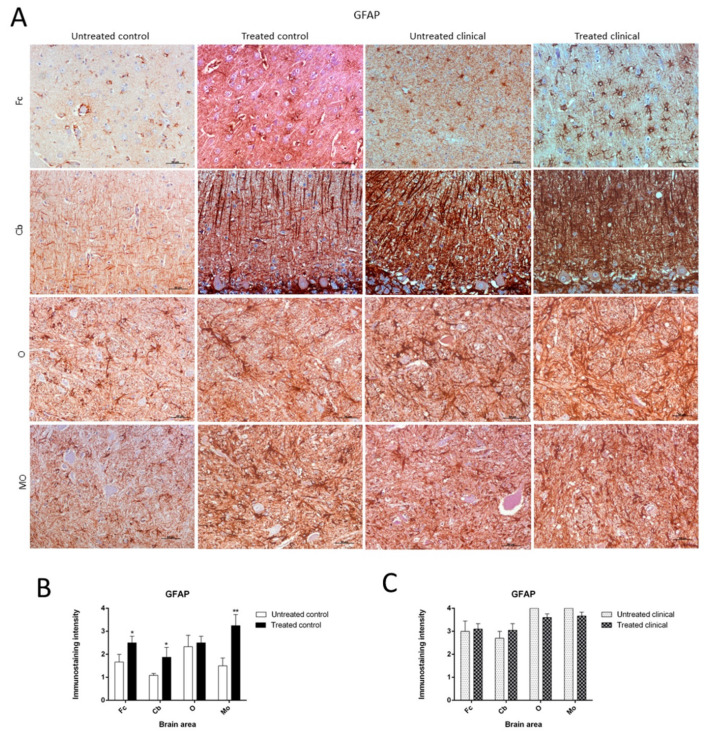
Glial fibrillary acidic protein (GFAP) immunostaining. (**A**) Increase of GFAP immunoreactivity in non-treated clinical scrapie compared with non-treated controls was evidenced. (**B**) Treated control animals showed significantly higher immunolabeling for GFAP than non-treated animals in all regions, except the obex. (**C**) GFAP immunostaining in the four regions examined did not reveal differences between treated and non-treated clinical animals. Medulla oblongata (MO), obex (O), cerebellum (Cb) and frontal cortex (Fc).

**Figure 6 ijms-21-03231-f006:**
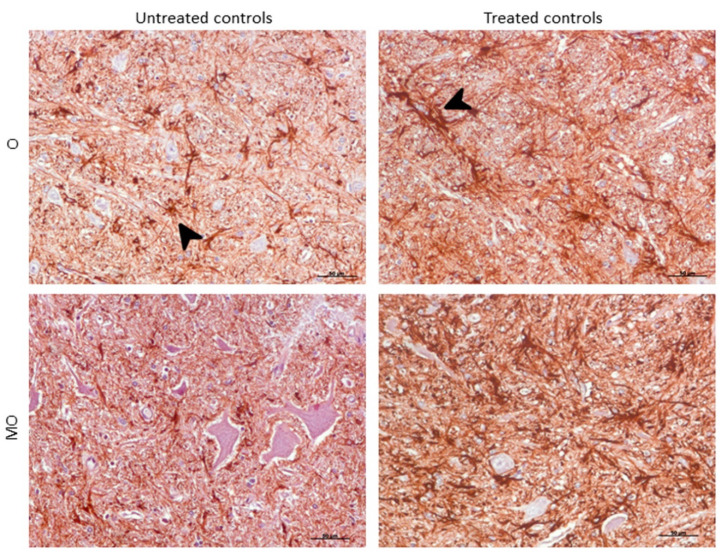
Morphological findings by GFAP IHC. Astrocytes in the medulla oblongata (MO) and obex (O) were widespread hypertrophic (arrowhead) after DEX treatment in controls compared to astrocytes in the non-treated ones, which appeared to be ramified (arrowhead).

**Figure 7 ijms-21-03231-f007:**
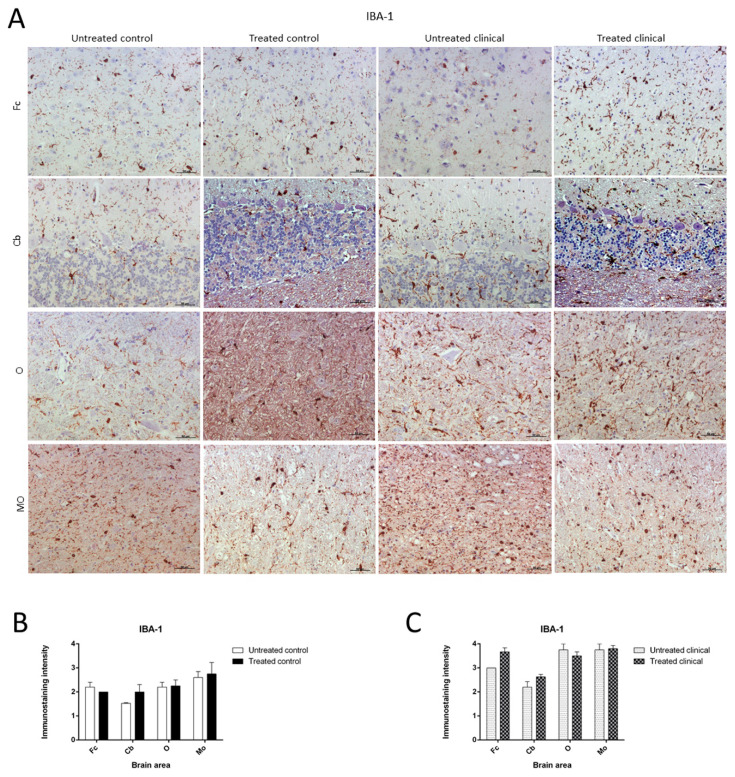
Ionized calcium-binding adaptor molecule-1 (IBA-1) immunostaining. (**A**) Microglial immunostaining intensity did not change with DEX treatment in the control and clinical groups. (**B,C**) No statistically significant differences in IBA-1 intensity between treated and non-treated animals were found. Medulla oblongata (MO), obex (O), cerebellum (Cb) and frontal cortex (Fc).

**Figure 8 ijms-21-03231-f008:**
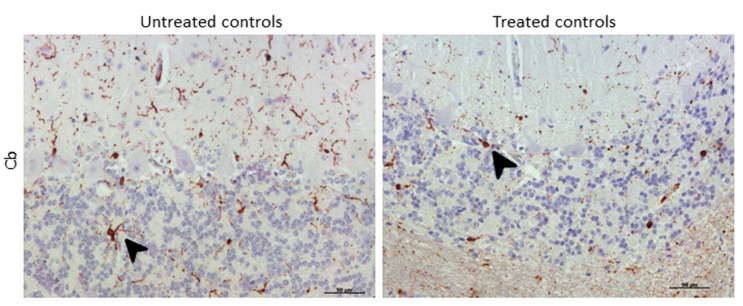
Morphological findings by IBA-1 IHC. A higher proportion of microglial cells in cerebellum presented an amoeboid (arrowhead) phenotype in DEX-treated controls compared to that in non-treated ones, which appeared to be mostly ramified (arrowhead).

**Figure 9 ijms-21-03231-f009:**
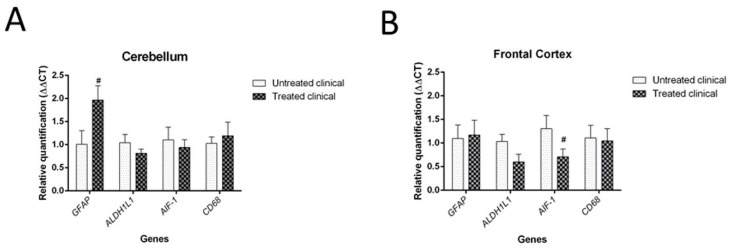
Results of gene expression by RT-qPCR in clinical individuals. (**A**) DEX treatment evidenced a tendency (#) toward increased GFAP mRNA levels in the cerebellum of treated animals compared with non-treated animals, although it did not reach statistical significance (*p* = 0.064). (**B**) Although it was not statistically significant (*p* = 0.085), there was a tendency toward decreased AIF-1 gene expression (microglia) in the frontal cortex of treated animals compared with that of animals in the non-treated group.

**Figure 10 ijms-21-03231-f010:**
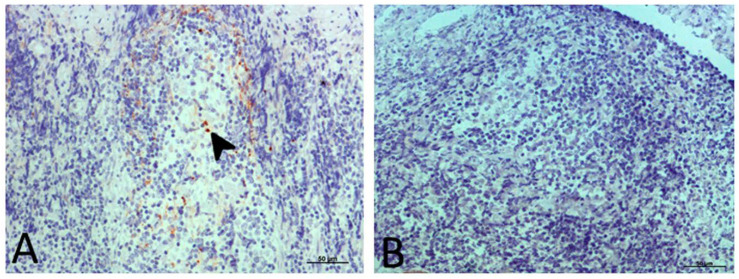
Immunostaining against pathological prion protein (PrP^sc^, arrowhead) with L42 antibody in biopsies of recto-anal mucosa-associated lymphoid tissue (RAMALT). (**A**) Scrapie-affected animal. (**B**) Control animal.

**Table 1 ijms-21-03231-t001:** Main clinical signs related to scrapie developed by both dexamethasone (DEX)-treated and non-treated clinical sheep.

Group	Sheep No	Main Clinical Signs
Clinical non-treated	11	Tremors, pruritus, ataxia, lost look
	12	Pruritus with skin lesions, alopecia, hyperexcitation
	13	Pruritus, alopecia, prostration
	14	Tremors, alopecia, prostration
	15	Pruritus
Clinical DEX-treated	16	Pruritus, ataxia, tremors
	17	Pruritus, alopecia
	18	Tremors, ataxia, hyperexcitation
	19	Pruritus, alopecia, hyperexcitation
	20	Tremors, intense widespread alopecia, prostration
	21	Hyperexcitation, prostration
	22	Tremors, constant pruritus, bruxism
	23	Scarce pruritus
	24	Intense pruritus with skin lesions
	25	Pruritus, alopecia, prostration

**Table 2 ijms-21-03231-t002:** Summary of data corresponding to animals included in the study.

Sheep No	*PRNP* Genotype	Age (Years)	Group	Treatment and Duration
**1**	ARQ/ARQ	9	Control	Untreated (13 months)
**2**	ARQ/ARH	10	Control	Untreated (16 months)
**3**	ARQ/ARQ	7	Control	Untreated (17 months)
**4**	ARQ/ARQ	4	Control	Untreated (17 months)
**5**	ARQ/ARQ	4	Control	Untreated (16 months)
**6**	ARQ/ARQ	8	Control	Untreated (9.5 months)
**7**	ARQ/ARQ	8	Control	Treated (10 months)
**8**	ARQ/ARQ	5	Control	Treated (16 months)
**9**	ARQ/ARQ	8	Control	Treated (16 months)
**10**	ARQ/ARQ	4	Control	Treated (16 months)
**11**	ARQ/ARQ	9	Clinical	Untreated (2 months)
**12**	ARQ/ARQ	4	Clinical	Untreated (<1 month)
**13**	ARQ/ARQ	5	Clinical	Untreated (1.5 months)
**14**	ARQ/ARQ	4	Clinical	Untreated (<1 month)
**15**	ARQ/ARQ	5	Clinical	Untreated (2.5 months)
**16**	ARQ/ARQ	6	Clinical	Treated (<1 month)
**17**	ARQ/ARQ	4	Clinical	Treated (<1 month)
**18**	ARQ/ARQ	4	Clinical	Treated (<1 month)
**19**	ARQ/ARQ	4	Clinical	Treated (<1 month)
**20**	ARQ/ARQ	5	Clinical	Treated (1 month)
**21**	ARQ/ARQ	4	Clinical	Treated (2 months)
**22**	ARQ/ARQ	7	Clinical	Treated (2 months)
**23**	ARQ/ARQ	4	Clinical	Treated (2 months)
**24**	ARQ/ARQ	4	Clinical	Treated (2 months)
**25**	ARQ/ARQ	4	Clinical	Treated (5 months)

**Table 3 ijms-21-03231-t003:** Primary specific antibodies used for immunohistochemical techniques and retrieval treatment applied for each antibody.

Antibody	Antigen	Type	Dilution	Retrieval Method	Source
L42	PrP^sc^	Monoclonal	1:500	Formic acid, 15 min Proteinase K, 15 min Heat treatment, 20 min Peroxidase blocking	DAKO
Anti-GFAP	GFAP	Polyclonal	1:500	Peroxidase blocking	DAKO
Anti-IBA-1	IBA-1	Polyclonal	1:1000	Heat treatment, 20 min Peroxidase blocking	WAKO

**Table 4 ijms-21-03231-t004:** TaqMan probes used for the RT-qPCR analysis.

Gene	Full Name	Reference	Source
**AIF-1**	Allograft inflammatory factor-1	Oa03222904_g1	Thermo Fisher
**ALDH1L1**	Aldehyde dehydrogenase 1 family member L1	Oa03267152_m1	Thermo Fisher
**CD68**	CD68 molecule	Oa04741636_g1	Thermo Fisher
**GFAP**	Glial fibrillary acidic protein	Oa03251662_m1	Thermo Fisher
**GUS-β**	β-Glucuronidase (reference gene)	Oa04828868_m1	Thermo Fisher
**HPRT-1**	Hypoxanthine phosphoribosyl transferase-1 (reference gene)	Oa04825272_gH	Thermo Fisher

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
