# Peer review of "Assessment of Glial Activation Response in the Progress of Natural Scrapie after Chronic Dexamethasone Treatment"

_ijms, 2020, doi:10.3390/ijms21093231_

Round 1
Reviewer 1 Report
The study is well designed, rigorously performed and executed and very well written. Overall, the study will be of high interest to study of neurodegenerative diseases.
Minor comment:
Synthetic GC dexamethasone (DEX) induce rapid degranulation of mast cells (important cells of neuroinflammatory pathogenesis), which release proinflammatory molecules promoting activation of microglia and astrocytes.
The increase of GFAP mRNA levels and a decrease of AIF1 gene expression may show an effect of a chain reaction, where DEX induces an autocrine / paracrine cell signaling in which mast cells first followed by activation of microglias. Consequently, reduction of activation of microglial cells, decrease of AIF1 gene expression and an increase of astrocyte activation.
The authors may acknowledge this possibility in the conclusion section and they may want to include a study in short and long term treatment of mast cells and glias cells.
Reviewer 2 Report
The manuscript is highly descriptive, although it is well written and contributes to the characterization of a scrapie model. However, before being considered for publication some issues should be addressed.
Because of the limitations in the approach used by the authors, the research question proposed was not resolved. Due to that, this study was short in determining the characteristics of the host immune response mediated by DEX in the scrapie model by indeed characterizing the role on glial cells in mediating this inflammatory-mediated immune response. I suggest modifying the title by emphasizing more the effect of DEX on glial cell activation more than in the immune response or include the appropriate experimental approach that allows to determination of cytokines profile by qRT-PCR, ELISA or flow cytometry.
Specific Comments:
a) The study could also be fortified by assessing if DEX can have a neuroprotective effect by evaluating the number of neurons using the NeuN marker. In addition, evaluating protein expression by western blotting can have a major impact on the conclusions obtained from this study.
b) How can the authors explain that treated control animals with DEX showed significantly higher GFAP immunolabelling than the non-treated group?
c) In both clinical groups, did the authors considerer the differences related to senescence processes associated with the age? Or related to the time of the treatment? How did you handle these variables?
d) The authors declared that their results show a downregulation of astrogliosis, reflecting “astroglial paralysis” during the clinical stage of scrapie. However, animals were treated for different periods of time, while controls were treated from 10 to 16 months, clinical groups ≤ 1 to 6 months. Can the duration of the treatment affect the interpretation made by authors about the “astroglial paralysis”? the prolonged DEX treatment effect?
e) Authors should improve the quality of the pictures, some are too pixelated and out of focus. Adding arrowheads or markers to highlight the changes in the pictures will be really useful for readers.
Reviewer 3 Report
This study aims to describe the potential modulation of neuroinflammatory responses during natural prion infection using the synthetic glucocorticoid dexamethasone. The authors discovered that DEX treatment of control animals elicited astrocyte response in GFAP, however DEX treatment of infected animals had minimal effect. In general this is a sound and well performed study that has been accurately described. There are some minor issues with the presentation of the text detailed below and also some suggestions to the methodology which may clarify the results and understanding of the study.
Within the text;
Introduction Line 45 where the authors introduce 'prion-like' grouping of diseases the text reads that due to similarities in neuropathology they were grouped and named prion like. This is misleading as it has long been known that neuropathology is similar across all these diseases. the hypothesis put forward was that the mechanism of induction and spread of protein misfolding within this group of diseases warrants their inclusion into a 'prion-like' grouping. This should be clarified.
Materials and methods line 185 the accepted nomenclature for RT-PCR is reference gene - the additional statement "as housekeeping or" is redundant and can be removed. - Also in Table 3 there is and e missing from both (reference gen)
Results & Methodology. The author had minor issues with the methodology compounded by the results presented. Throughout this article analysis has been performed by subjective scoring of vacuolation or IHC staining by two independent observers. The compound scores are averaged and presented as bar graphs underneath images which appear not to support them.
I would suggest rather than subjective scoring, non-subjective IHC image analysis to accurately quantify the amount (% area x intensity*) of staining in each given brain area should be performed. Data should be presented as dot plots so inter-animal variation and N are accurately depicted not bar charts of group means +/-SEM. *this type of analysis can be performed to a limited extent on DAB stained images, confirmation by non-amplifying immunfluorescent IHC would be preferred.
The eye is drawn to Cb treated clinical case as an example with the strongest staining in Figure 5 not confirmed by bar chart beneath.
In Figure 6 GFAP staining in O and MO (? the names of this region should be fully described either on the figure or within the legend for clarity) of treated control, untreated clinical and treated clinical look identical yet are reported underneath as 1 score lower in untreated for both regions? Again Cb region appears more intense in treated vs untreated clinical not supported by graph below - yet is supported by gene expression data in Figure 10.
In Figure 7 could the statement 'vast majority' be replaced with 'widespread' - no effort has been made to quantify total astrocytes or numbers in each state, plus immunostaining is only via 1 marker (GFAP)
Unfortunately this highlights one of the major limitations of this study, the reliance on single antibody/marker to characterise each cell type gives very little information other than morphology and abundance.
GFAP is a pan-reactive marker of astrocytes (ref 107) so it is impossible to determine if DEX treatment in uninfected state has induced A1 or A2 like reactivity or how this impacts or not with prion infection. This could potentially have been addressed by RT-qPCR for A1/A2 markers?
Given proper image analysis this would constitute a fairly basic description of what the authors intended to achieve.
Round 2
Reviewer 2 Report
The authors included some of the suggestions in the manuscript including modifying title and the comments that allow clarifying some of my concerns in the discussion. In addition, they improved the pictures and complete the statistical information.